# Blood Loss Following Open Posterior Spinal Fusion in Fractures: Cannulated vs. Solid Pedicle Screws

**DOI:** 10.3390/jpm13010160

**Published:** 2023-01-16

**Authors:** Pedram Rajabifard, John Edward Cunningham, Michael A. Johnson, Henrik Constantin Bäcker, Peter Turner

**Affiliations:** 1Department of Orthopaedic Surgery, Royal Melbourne Hospital, 300 Grattan Street, Parkville, VIC 3050, Australia; 2Department of Orthopaedic Surgery, Epworth Richmond Hospital, 89 Bridge Road, Richmond, VIC 3121, Australia; 3Centre for Musculoskeletal Surgery, Charite Berlin, Chariteplatz 1, 10117 Berlin, Germany

**Keywords:** posterior spinal fusion, blood loss, blood transfusion, pedicle screws

## Abstract

We aim to delineate whether there is increased blood loss with the use of cannulated pedicle screws compared to solid pedicle screws in patients undergoing posterior spinal fusion. A single-centre retrospective case-control study was undertaken on patients undergoing PSF for spinal fractures. Cannulated screw fixation was compared with solid screw fixation. Intraoperative blood loss was estimated using pre and postoperative haemoglobin levels, recorded estimated blood loss and cell saver reports. Anticoagulation, blood product administration, operative time and number of levels fused were assessed. A total of 64 cases, 32 in each cohort, were included in the analysis. Overall mean haemoglobin reduction from pre- to post-operative was 2.82 ± 1.85 g/L per screw inserted in the cannulated group, compared to a haemoglobin decrease of 2.81 ± 1.521 g/L per screw inserted in the solid screw group (*p* = 0.971). Total estimated intraoperative blood loss was 616.3 + 355.4 mL in the cannulated group, compared to 713.6 + 473.5 mL in the solid screw group (*p* = 0.456). Patients with preoperative thrombocytopenia had a transfusion rate of 0.5 ± 0.71 units/patient compared to 0.04 ± 0.19 units/patient in patients with normal platelet levels (*p* < 0.005). The differences in blood loss observed between cannulated and solid pedicle screws are non-significant overall. The largest predictor for need of transfusion was pre-operative thrombocytopenia, regardless of the type of screw used.

## 1. Introduction

Spinal fractures are an increasing orthopaedic problem, with an overall incidence quoted to be up to 89 per 100,000 population. Of those with spinal fractures, between 16 and 64 per 100,000 population need to be hospitalized for their injuries. Men are predominantly affected, accounting for up to 60% of patients [1,2]. These injuries are commonly traumatic, with falls in the elderly and motor vehicle accidents in the young accounting for the majority of the mechanism of injury.

The decision of the management method, non-operative or operative, for spinal fractures remains a topic of controversy [3,4]. Posterior spinal fusion with pedicle screw fixation remains an accepted mode of operative fixation in thoracic and lumbar spine instability caused by traumatic fractures [4]. The goal of fixation is to correct deformity and provide stabilisation until fusion has occurred, while also providing decompression of the canal [5]. The pedicle is the strongest posteriorly accessible site for instrumentation, with pedicle screw fixation first being introduced by Boucher in the 1950s.

Many technological and design advances in pedicle screws have occurred since their inception, including but not limited to the production of cannulated pedicle screws as an alternative to the traditional solid pedicle screw. Cannulated pedicle screws, unlike their solid counterparts, are advantageous as they can be used for both percutaneous and open procedures, as well as having an option for cement augmentation through the cannulated lumen. These additional capabilities mean that cannulated pedicle screws are a more versatile implant; cannulated screws can therefore help to reduce inventory costs. Cannulated pedicle screws are increasing in popularity, with 52% of all pedicle screws being cannulated in 2016 in the United States [6].

One of the major surgical risks of posterior spinal fusion is blood loss. Increased blood loss is associated with increased transfusion requirements and prolonged inpatient stays which may be due to surgical site infection, transmission of infection from blood products or fluid shifts resulting in cardiopulmonary compromise [1]. Open procedures such as those with solid pedicle screws are postulated to have increased blood loss compared to minimally invasive surgical techniques [7] such as percutaneous posterior fusion with cannulated pedicle screws, due to the increased exposure leading to increased bleeding points from well perfused bone [1]. This hypothesis for reduced intraoperative blood loss with cannulated screws therefore does not relate to prosthesis design, but rather a surgical approach.

Studies have shown that, without preventative strategies to reduce intraoperative haemorrhage, transfusion rates can be as high as 81% [8]. While some strategies to reduce intra-operative blood loss or the need for blood transfusion such as administration of tranexamic acid have mounting support for their use [9,10,11,12], other strategies such as permissive hypotension have inconsistent evidence [8].

The primary aim of this paper is to determine whether the use of cannulated pedicle screws is associated with increased intraoperative blood loss in patients undergoing thoracic and lumbar posterior spinal fusion in patients suffering from traumatic vertebral column fractures. Our secondary aim was to determine other factors which may influence intraoperative blood loss or blood transfusion requirements.

## 2. Materials and Methods

A single-centre retrospective case-control study of patients undergoing posterior spinal fusion of the thoracic and/or lumbar spine for traumatic fracture was undertaken at a major level 1 trauma centre. Ethical approval was obtained from our local international review board.

Electronic medical records were used to identify patients who had undergone posterior spinal fusion for management of traumatic fracture(s). Inclusion criteria were patients over the age of 18 years who underwent posterior spinal fusion for thoracic and/or lumbar spine fractures sustained in a traumatic event. Exclusion criteria were patients who were on therapeutic anticoagulation prior to their posterior spinal fusion, those who underwent concurrent surgeries at the time of their posterior spinal fusion and patients who were not haemodynamically stable as assessed by the anaesthetic and orthopaedic team prior to undergoing their posterior spinal fusion.

A control group of patients who underwent posterior spinal fusion using a cannulated pedicle screw system (Everest MI, Lifehealthcare, North Ryde, New South Wales, Australia) were selected. Diagnoses of each patients’ injuries were determined by obtaining the operation and admission notes and pre-operative imaging (computed tomography and/or magnetic resonance imaging of the thoracic and/or lumbar spine). For injury classification, the AO Spine Thoracolumbar Injury Classification system was used.

Based on the operation and clinical notes and post-operative imaging, the number of levels fused and the prostheses utilised were determined. This included the number of pedicle screws placed (including whether there were pedicle screws inserted at the fracture site), and whether there was utilisation of a sublaminar hook or rod crosslink.

These patients were then matched to a group of patients who had undergone posterior spinal fusion using a solid pedicle screw system (Reline, Nuvasive, San Diego, CA, USA), using the same process of identification noted above. The patient groups were matched to age +/- 10 years and number of screws implanted +/- 1 screw.

Intraoperative blood loss was assessed using a variety of methods. For all patients, preoperative and post operative haemoglobin levels were noted as a means of assessing intraoperative blood losses. Where multiple haemoglobin levels were available for use, the results with the closest temporal relationship were utilised (i.e., the haemoglobin level taken closest to the start of the operation time as the pre-operative datapoint, and the first post operative haemoglobin level taken). Additionally, note was made of intraoperative blood loss estimates and cell salvage reports as a means of assessing intraoperative blood losses. This included volume of blood reinfused using cell salvage.

Anaesthetic reports were used to identify operative time, tranexamic acid use, volume of intraoperative crystalloid infusion and intraoperative transfusion of blood products. It was also noted if any other surgical interventions were carried out at the same time as the posterior spinal fusion.

Factors that may affect blood loss were noted, including pre-operative platelet levels and coagulation studies, pre-injury use of anticoagulants as well as chemical prophylaxis of venous thromboembolism with low molecular weight heparins both prior and post posterior spinal fusion surgery.

For statistical analysis, Excel spreadsheet (Microsoft Excel for Mac, v16.69, Microsoft Corporation, Redmond, Washington, DC, USA) and IBM SPSS v28 (IBM Company, Chicago, IL, USA) were used to evaluate 2 tailed *t*-tests. Normally distributed variables are presented as a mean and standard deviation. The level of significance was set to a *p*-value of ≤0.05.

## 3. Results

A total of 64 cases, 32 cannulated and 32 solid screws fit the inclusion and exclusion criteria and were included in the analysis. These 64 patients underwent a total of 64 posterior spinal fusion surgeries, completed by a total of four fellowship trained spine surgeons who underwent the same training. All patients underwent a general anaesthetic with an open approach; and no patient had a drain placed. Most surgeries were performed using a reinfusion system, cell saver, with reinfusion occurring during the intraoperative period.

Table 1 outlines the major findings of the study. The two groups were homogenous with regard to age, number of screws inserted, and number of levels fused. The average age of patients was 41.3 ± 15.4 years in the cannulated screw group compared to 41.1 ± 15.7 years in the solid screw group (*p* = 0.473). The cannulated pedicle screw cohort was comprised of 30 men (86%) while the solid pedicle screw cohort was comprised of 27 men (77%). A median of 4.14 ± 1.52 levels were fused in the cannulated group, compared to 4.2 ± 1.49 in the solid screw group (*p* = 0.437). An average of 9.03 ± 2.5 and 9.03 ± 2.6 screws were inserted per case in each group, respectively (*p* = 0.500).

There were patients in both groups who had concomitant injuries as a result of their mechanism of injury. Due to the heterogeneity of the injury patterns, other injuries were not considered in this study. There were, however, three patients, all in the solid screw cohort, who had concurrent surgeries at the time of their posterior spinal fusion. One patient had a one level anterior cervical discectomy and fusion for C5 vertebral body fracture, one patient had a tibial intramedullary nail fixation with fasciotomies for an open tibia fracture, and one patient had an ankle and talus open reduction internal fixation for open talar fracture.

Out of the seventy patients reviewed, only one patient was noted to be taking aspirin pre-operatively. This patient was part of the cannulated pedicle screw group and was using aspirin as secondary prevention for ischaemic heart disease. There were no patients identified as taking any other antiplatelet or anticoagulant medication pre-operatively in either group.

Overall haemoglobin reduction from pre- to post-operative was 23.4 ± 11.3 g/L versus 23.9 ± 11.6 g/L (*p* = 0.862). This was extrapolated to a mean haemoglobin decrease of 2.82 ± 1.85 g/L per screw inserted in the cannulated group, compared to a haemoglobin decrease of 2.81 ± 1.52 g/L per screw inserted in the solid screw group (*p* = 0.971) (Table 1).

Comparing fractures in the thoracic spine to fractures in the lumbar spine did not result in any statistically significant differences. For fractures of the thoracic spine, the average haemoglobin drop per screw was 1.79 ± 0.76 g/L in the cannulated cohort compared to 2.34 ± 1.25 g/L in the solid screw cohort (*p* = 0.165). For fractures of the lumbar spine, the average haemoglobin drop per screw in the cannulataed cohort was 3.24 ± 2.06 g/L compared to 3.35 ± 1.68 g/L in the solid screw cohort (*p* = 0.680).

Total estimated intraoperative blood loss was 616.3 ± 355.4 mL in the cannulated group, compared to 713.6 ± 473.5 mL in the solid screw group (*p* = 0.456) (Table 1).

Average operating time was 111.8 ± 36.7 min in the cannulated group, compared to 111.4 ± 32.3 in the solid screw group (*p* = 0.962) (Table 1).

When analysing blood loss per unit time, the average blood loss per hour in the cannulated group was calculated to be 338.0 + 198.7 mL/hour compared to 331.66.5 ± 197.3 mL/h in the solid pedicle screw group (*p* = 0.643) (Table 1).

There were five patients in each group who were found to be thrombocytopenic preoperatively, defined as a platelet level of less than 150 × 109/L (*p* < 0.01). In the cannulated pedicle screw group, 2 out of the 5 patients who were thrombocytopenic pre-operatively required intraoperative blood transfusion, each with one unit of packed red blood cells (*p* = 0.009). This transfusion rate was much higher than in those patients who had normal platelet levels pre-operatively, of whom only 1 out of a total of 27 patients received intra-operative blood transfusions. Similarly, in the solid pedicle screw group, 2 of the 5 patients who were thrombocytopenic pre-operatively received intra-operative blood transfusions, with up to two units transfused. Of the remaining 27 patients with normal platelet levels, only one patient required an intraoperative blood transfusion. All 10 thrombocytopenic patients received 1000 mg of tranexamic acid intravenously at anaesthetic induction.

Overall, this led to a transfusion rate of 0.5 ± 0.71 units/patient in patients with thrombocytopenia compared to a transfusion rate of 0.04 ± 0.19 units/patient in patients with normal platelet levels (*p* < 0.005).

Interestingly, however, patients who were thrombocytopenic prior to their operation had an average haemoglobin drop of 1.70 ± 1.25 g/L per screw inserted, compared to an average drop of 3.02 ± 1.67 g/L per screw inserted in patients with normal pre-operative platelet levels (*p* = 0.02).

## 4. Discussion

To our knowledge, there are no studies that compare blood loss between the cannulated and solid pedicle screw types in open posterior spinal fusion surgery. This study showed that undertaking open posterior spinal fusion with cannulated pedicle screws was not associated with any statistically significant change in intraoperative estimated total blood loss compared to using solid pedicle screws. Similarly, there was no statistically significant difference found with respect to Cell Saver blood loss, amounts of intraoperative fluid and/or blood transfusion or changes in haemoglobin levels.

When undertaking posterior spinal fusion for traumatic thoracolumbar fractures, the surgeon has a choice in using either cannulated or solid pedicle screws. Cannulated pedicle screws have multiple purported advantages when compared with solid pedicle screws. They exhibit surgical versatility, are able to be used in both open and minimally invasive, or percutaneous, spinal procedures. Their ability to be used in multiple surgical approaches means that, from an economic standpoint, they would be the preferred stocked item at a hospital. Additionally, they have been associated with reduced post-operative muscle pain, infection risk and improved cosmesis when used in a minimally invasive fashion [13]. The gravity of these advantages must be weighed against any disadvantages that may arise from their use.

Blood loss in spinal surgery is a major risk to be mitigated, with spinal surgery ranking among the top surgical procedures associated with need for blood transfusions [7]. It is no surprise that increased blood loss can lead to poor outcomes such as prolonged inpatient stay, increased transfusion requirements and infection rate [1]. It is therefore imperative to determine the factors that increase intraoperative bleeding and thus transfusion requirements in spinal surgery [7].

While literature supports that cannulated screws are associated with reduced blood loss in posterior fusion [13], this is typically attributed to the surgical approach resulting in reduced soft tissue damage and not the implant design [14]. Cannulated screws are most utilised in minimally invasive, percutaneous procedures, distinct to our centre which uses them in an open approach. Undertaking posterior spinal fusion in an open manner has been shown to have a significant increase in risk in transfusion requirements [7].

Our analysis showed that, when calculating blood loss per unit time, there was a non-significant slight reduction in blood loss per hour of operative time in patients who had posterior fusion with cannulated pedicle screws, with a comparable total operative time in both groups.

A subgroup analysis comparing patients who had pedicle screws inserted at the fracture site compared to those who had no screws at the fracture site failed to show any statistically significant difference in the total estimated intraoperative blood loss volume. Breaking this down further into cannulated and solid screw groups showed that the patients who received cannulated pedicle screws at a fracture site had a smaller reduction in haemoglobin per screw inserted compared to those who did not have screws inserted at the fracture site, but again this was not found to be statistically significant.

This finding was in keeping with current literature results, which show that inserting pedicle screws at the fracture site was not associated with an increase in intraoperative bleeding [15]. Given the mounting evidence that insertion of screws at the fracture site increases construct biomechanical strength [15,16,17], our research would support that this practice is safe with regard to blood loss volumes.

Our analysis showed that the factor most closely correlated with increased transfusion requirement was pre-operative thrombocytopenia. Trauma is a common cause of thrombocytopenia [18] and therefore thrombocytopenia is not unexpected in our patient group. Despite the thrombocytopenic patients showing increased transfusion requirements, they also had a smaller reduction in haemoglobin levels post-operatively. This incongruence is likely related to the confounding factor that intraoperative blood transfusion would elevate the post-operative haemoglobin. With such a significantly higher transfusion rate in the thrombocytopenic patient population, surgical teams should be acutely aware of the increased transfusion risks and therefore more closely monitor these patients both in the intra- and post-operative period.

Our results showed no statistically significant difference when comparing reduction in haemoglobin in patients who received prophylactic chemical deep vein thrombosis (DVT) prophylaxis, defined as weight-adjusted subcutaneous enoxaparin, administered within 24 h of their operation start time compared to patients who did not receive prophylactic chemical DVT prophylaxis, regardless of whether the patient had cannulated or solid pedicle screws placed.

As the rate of use of cannulated pedicle screws increases, another surgical factor important to analyse is operative time. Despite an initial learning curve being associated with implanting cannulated pedicle screws, their use in this study was not associated with any statistically significant change in operative time between cannulated and solid pedicle screws, whether this was analysed in terms of the operation time overall or operative time per screw inserted. This can perhaps be explained by the fact that there is little difference in surgical technique between using cannulated and solid pedicle screws when undertaking open posterior spinal fusion. This explanation may not be able to be extrapolated to minimally invasive surgical approaches, as the surgical technique for implantation of cannulated screws in such a manner has significant differences compared to the open approach [13].

Limitations of this study include small cohort, single centre study and its retrospective nature. In our centre, we have switched from the use of solid pedicle screws to the use of cannulated pedicle screws for all patients; therefore, conducting anything but a retrospective analysis on differences between cannulated and solid pedicle screw use is difficult. Ideally, further studies with a larger cohort obtained from more surgeons and surgical centres are undertaken using higher powered study designs to more definitively determine differences in clinical outcomes between cannulated and solid pedicle screw fixation constructs.

Post-operative haemoglobin levels were generally taken within 24 h of completion of the posterior spinal fusion. While patients in this study were haemodynamically stable at this time, there may still be ongoing fluid shifts which would later be reflected in the full blood count [19].

Some of the patients included in this study had concomitant injuries sustained in their trauma. Despite being haemodynamically stable at time of their posterior spinal fusion, these injuries could still act as confounders in changing haemoglobin levels. Further studies should aim to only include patients with a single diagnosis as part of the study.

## 5. Conclusions

The use of cannulated screws has economical advantages due to their versatility in being able to be used for percutaneous and open procedures. Although they are postulated to be associated with increased blood loss compared to their solid counterparts due to the loss of blood through the cannulated lumen, our study showed no statistically significant difference in blood loss when cannulated pedicle screws are used in comparison with solid pedicle screws for open posterior spinal fusion following spinal trauma. The factor most closely correlated with intraoperative blood transfusions was pre-operative thrombocytopenia. Clinicians should be aware of this risk and increase their monitoring of these patients.

## Figures and Tables

**Table 1 jpm-13-00160-t001:** Major findings.

	Cannulated Screws	Solid Screws	*p*-Value
Number	32	32	
Gender (male)	27	25	0.529
Age (years)	42.2 + 15.7	42.1 + 15.8	0.993
Number of screws	9.19 + 2.5	9.19 + 2.6	1
Number of levels fixation	4.12 + 1.60	4.19 + 1.60	0.876
Blood loss (Cell saver) (mL)	257.2 + 211.4	255.5 + 179.7	0.385
Total blood loss (mL)	616.3 + 355.4	713.6 + 473.5	0.456
Hb difference (g/L)	23.4 + 11.3	23.9 + 11.6	0.862
Intraoperative fluids (mL)	1453.1 + 836.3	1723.4 + 914.8	0.222
Blood loss (Cell Saver) per screw (mL/screw)	26.5 + 21.0	30.3 + 24.3	0.561
Total blood loss per screw (mL/screw)	67.0 + 38.7	58.3 + 47.9	0.521
Hb preoperative (g/L)	132.0 + 23.7	131.3 + 21.7	0.908
Hb postoperative (g/L)	108.5 + 23.7	107.4 + 17.0	0.804
Hb loss per screw (g/L)	2.82 ± 1.85	2.81 ± 1.52	0.971
Intraoperative transfusion	0.09 + 0.30	0.13 + 0.42	0.732
Operative time (min)	111.8 + 36.7	111.4 + 32.3	0.962
Total blood loss per hour (mL/h)	338.0 + 198.7	366.5 + 197.3	0.643

## Data Availability

The data presented in this study are available on request from the corresponding author. The data are not publicly available due to privacy reasons.

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
