# Peer review of "Blood Loss Following Open Posterior Spinal Fusion in Fractures: Cannulated vs. Solid Pedicle Screws"

_jpm, 2023, doi:10.3390/jpm13010160_

Round 1
Reviewer 1 Report
Were the patient with other injuries (i.e other fractures, internal organs injuries) excluded from the assessment?
Did You consider checking the difference in blood loss according to the level of fracture (thoracic vs lumbar)?
Author Response
Thank you for your review of our paper.
- There were patients with other injuries, however, the injuries were too heterogenous for inclusion in this study. I have made a note of this in the paper in the results and limitations sections.
- I have checked the results between thoracic and lumbar fractures in the two groups and noted the subgroup analysis in the results section.
Reviewer 2 Report
This manuscript described a retrospective clinical study comparing the blood loss of cannulated vs solid pedicle screws for spinal fracture surgery fixation. The results showed the newer cannulated pedicle screws did not cause more blood loss and revealed that pre-operative thrombocytopenia was a factor that determined the need for blood transfusion. Overall, the manuscript is well written. I have the following comments that need to be addressed.
1. Lines 199 and 211, “reduced reduction” changed to “smaller reduction or less reduction”.
2. The authors only have one results data table, when describing the results, the table should be indicated where appropriate.
3. Conclusion: Two paragraphs should be combined and the tune of the conclusion should be changed like below:
“The use of cannulated screws has economical advantages due to their versatility in being able to be used for percutaneous and open procedures. Although they are postulated to be associated with increased blood loss compared to their solid counterparts due to the loss of blood through the cannulated lumen, our study showed no statistically significant difference in blood loss when cannulated pedicle screws are used in comparison with solid pedicle screws for open posterior spinal fusion following spinal trauma. The factor most closely correlated with intraoperative blood transfusions was pre-operative thrombocytopenia. Clinicians should be aware of this risk and increase their monitoring of these patients”.
Author Response
Thank you for your review of our paper.
- I have made these adjustments
- I have referred to Table 1 appropriately throughout the text. I have also expanded on table 1
- I have altered the conclusion as suggested